# The Dynamics of Hepatic Fibrosis Related to Schistosomiasis and Its Risk Factors in a Cohort of China

**DOI:** 10.3390/pathogens10121532

**Published:** 2021-11-23

**Authors:** Fei Hu, Shu-Ying Xie, Min Yuan, Yi-Feng Li, Zhao-Jun Li, Zhu-Lu Gao, Wei-Ming Lan, Yue-Ming Liu, Jing Xu, Dan-Dan Lin

**Affiliations:** 1Jiangxi Provincial Institute of Parasitic Diseases, Jiangxi Province Key Laboratory of Schistosomiasis Prevention and Control, Nanchang 330096, China; hufei.nc@gmail.com (F.H.); xsy0317@163.com (S.-Y.X.); yuanmin2003iou@gmail.com (M.Y.); liyifeng1004@163.com (Y.-F.L.); 13576086889@163.com (Z.-J.L.); gzl63jx@163.com (Z.-L.G.); wmlan0795@163.com (W.-M.L.); lym610615@163.com (Y.-M.L.); 2National Institute of Parasitic Diseases, Chinese Center for Disease Control and Prevention, WHO Collaborating Centre for Tropical Diseases, Chinese Center for Tropical Disease Research, Shanghai 200025, China

**Keywords:** schistosomiasis, hepatic fibrosis, risk factors, dynamics

## Abstract

China has had a long history against schistosomiasis japonica. The most serious prognosis of chronic schistosome infection is hepatic fibrosis, which develops into advanced schistosomiasis if the process is not effectively controlled. After a more than seven decades endeavor, China has gained remarkable achievements in schistosomiasis control and achieved transmission control nationwide (infection rate of schistosomes in residents and domestic animals both less than 1%) by 2015. However, new advanced schistosomiasis cases emerge annually in China, even in areas where the transmission of schistosomiasis had been interrupted. In the present study, the residents (>5 years old) in a schistosomiasis endemic village were examined for schistosomiasis every year during 1995–2019 by the modified Kato–Katz thick smear method and/or miracidium hatching technique. Residents who were identified to have an active infection method were treated with praziquantel at a dose of 40 mg/kg body weight. Ultrasonography was carried out to assess the liver morbidity related to schistosomiasis in 1995 and 2019, respectively. The prevalence of schistosomiasis among residents presented a downward trend annually, from 17.89% (175/978) in 1995 to 0 (0/475) in 2019. Among 292 residents who received ultrasound scan both in 1995 and 2019, 141 (48.29%) presented stable liver damage, while liver fibrosis was developed severely in 86 (29.45%) and reversed in 65 (22.26%) residents. Univariate and multivariate analysis showed that anti-fibrosis treatment was the protective factor against schistosomiasis hepatic fibrosis. Males, residents aged 38 and above, fishermen, and people who did not receive anti-fibrosis treatment were groups with higher risk of liver fibrosis development. Our results revealed that although the infection rate of schistosome dropped significantly in endemic areas, liver fibrosis was still developing among some residents, even though they had received deworming treatment. Liver protection/anti-fibrosis treatment should be administered in endemic regions and regions with historically uncontrolled transmission to slow down the deterioration of hepatic fibrosis among patients in schistosomiasis endemic areas.

## 1. Introduction

Schistosomiasis, a water-borne disease, is endemic in 78 countries in Africa, South America, and Asia [1]. It is also a serious global health problem and is a neglected tropical disease [2]. In China, schistosomiasis japonica caused by *Schistosoma japonicum* was epidemic in 12 provinces (cities, autonomous regions) along the Yangtze River, leading to heavy disease burden [3]. For *S. japonicum*, eggs are the pivotal pathogenic factor, inducing an inflammatory cascade that includes the deposition of collagen and extracellular matrix proteins when they are transported into liver, thus leading to fibrosis when fibrogenesis exceeds the replacement of healthy cells with scar tissue. After more than seven decades of efforts and hard work, schistosomiasis has been effectively controlled in endemic areas through comprehensive interventions. By the end of 2015, China achieved the national criteria of transmission control (infection rate of schistosomes in residents and domestic animals less than 1%) [4]. There were 30,175 schistosomiasis cases documented in 2019, in which the majority were advanced cases with various clinical manifestations [5].

Chemotherapy with praziquantel plays an important role in decreasing morbidity and prevalence of schistosomiasis. However, liver parenchymal fibrosis (liver fibrosis, the same below) occurred in some cases even though they received standard treatment against schistosomes. Previous reports suggested that around 20% of patients would develop liver fibrosis [6]. In China, about 1000 new advanced schistosomiasis cases are documented annually, while some of these are reported from endemic areas where the transmission of schistosomiasis was interrupted many years ago [7]. Understanding the dynamics of liver fibrosis led by schistosomiasis and exploring related risk factors are required to design or implement effective interventions timely to prevent the occurrence or decrease the development of advanced cases.

Many factors are reported to affect the process of liver fibrosis, including occupation, education level, duration of water exposure, frequencies infected with schistosomes, times received treatment, etc., but most conclusions were based on single factor analysis [8,9,10,11,12,13,14]. In addition, considering that it often takes 5–15 years for liver function to shift from compensated stage to decompensated stage, periods of some research are too short to provide effective information [15]. In our study, multi-year cross sectional surveys on prevalence of schistosomiasis and ultrasound examination were conducted in a selected village in Jiangxi Province of China during 1995 and 2019, to understand the dynamic changes of liver fibrosis and explore its influence factors.

## 2. Results

### 2.1. General Information of Participants

The number of participants from the surveyed village ranged from 309 to 978 during 1995–2019, with the ratio of male to female in the range of 1:0.87 to 1:1.22. As the observation went on, the number of participants reduced year by year, with the average age increased (Figure 1). The main reason for the reduction of participants is that the residents flowed frequently because of migrant work, thus the compliance of residents gradually decreased year by year. In addition, there were also some residents who refused to be examined because they were no longer in contact with infested water due to the increase of age.

### 2.2. Changes of Schistosomiasis Prevalence in Residents

The prevalence of schistosomiasis among residents decreased from 17.89% (175/978) in 1995 to 0 (0/475) in 2019, presenting a downward trend year by year (Figure 2). In 2007, due to the change in the parasitological method, the prevalence of schistosomiasis rebounded to 8.29% (63/760), but without showing significant difference in prevalence between 2007 and 2004 (χ^2^ = 3.567, *p* = 0.059). In the following years, the prevalence of schistosomiasis dropped sharply and has been less than 1% since 2011. Meanwhile, the infection intensity (EPG, number of eggs per gram of feces) varied highly but was consistent with the tendency of the prevalence of schistosomiasis (*r* = 0.928, *p* = 0.000). The re-infection rates among residents ranged from 4.29% to 38.46% during 1996–2011. Since 2011, reinfection only occurred in 2012 and 2017 [16].

### 2.3. Changes of Hepatic Fibrosis Related to Schistosomiasis in Residents

Totally, 292 residents received ultrasound scan and fecal examination in 1995 and were followed up by same methods in 2019. The ratio of male to female was 1:1. In 1995, the males and females were at ages of 36.46 ± 12.59 and 36.69 ± 9.55, respectively. Among of them, hepatic fibrosis was classified as Grade 0 in 92 participants, while 38, 7, and 4 participants had hepatic fibrosis classified as Grade I, Grade II, and Grade III, respectively, in both years. Overall, the rate of hepatic fibrosis grading II or III was higher in 2019 than that in 1995. The grading of hepatic fibrosis was stable—as graded by the aforementioned scale in 48.29% of participants (141/292) over the observed years. Hepatic fibrosis was deteriorated in 29.45% (86/292) of participants but recovered in 22.26% (65/292) of residents, with significant difference detected (χ^2^ = 3.939, *p* = 0.047) (Table 1).

### 2.4. Factors Influencing the Evolution of Hepatic Fibrosis

Univariate logistic analysis showed that gender, age, education level, result of fecal examination, frequency of infested water contact, the number of previous treatments against schistosomiasis, and compliance of treatment were not correlated with the recovery of schistosomiasis liver fibrosis. However, factors such as occupation and anti-liver fibrosis treatment can affect the recovery of liver fibrosis. The recovery rate of liver fibrosis among fishermen (12.12%, 8/66) was significantly lower than that in non-fishermen (25.22%, 57/226) (χ^2^ = 5.066, *p* = 0.024), while the recovery rate of liver fibrosis (27.33%, 41/150) among residents who received anti-liver fibrosis treatment was significantly higher than that of those who did not receive anti-liver fibrosis treatment (16.90%, 24/142) (χ^2^ = 4.587, *p* = 0.032). In addition, male, elder people, fishermen, people who presented egg-positive in either 1995 or 2019, people received less treatment were at higher risk of deteriorating of hepatic fibrosis with OR values higher than 1 and *p* < 0.05 (Table 2).

The factors of *p* value < 0.2 in the results of the univariate analysis were included in the multivariate logistic analysis. The results showed that anti-fibrosis treatment is the protective factor against schistosomiasis hepatic fibrosis. Male, people aged 38 or above, fishermen, and people who did not receive anti-fibrosis treatment are at a higher risk of liver fibrosis deterioration (Table 3).

## 3. Discussion

Jiangxi is one of the most severe schistosomiasis-endemic provinces in China. To conquer schistosomiasis, intensive interventions were conducted in Jiangxi Province under the guidance of the national strategic plan of schistosomiasis control and prevention, resulting in great achievements in schistosomiasis control [17,18]. To evaluate the effectiveness of control programmes, conducting assessment of morbidity and prevalence is important using sensitive and accurate methods. With the decrease of the prevalence and infection intensity of schistosomiasis in residents, the sensitivity and accuracy of routine parasitological methods such as the modified Kato–Katz thick smear method were insufficient to find all schistosomiasis cases [19,20,21]. In our study, we collected more stool samples from participants, prepared and read more slides to overcome the shortcoming of the Kato–Katz method and then combined with the hatching method during the later period (2007–2015). Thus, it led to an abnormal rise of the positive rate from 5.83% in 2004 to 8.29% in 2007. Therefore, in order to achieve the goal of eliminating schistosomiasis by the goal of 2030 in China as scheduled, more sensitive detection techniques should be explored to detect patients, especially those with asymptotic and light infections. 

In the life cycle of *Schistosoma japonicum,* egg is the major pathogen causing granulomatous inflammation which often leads to chronic liver damage. Among those severe cases, owing to the deposition of egg granuloma and fibrosis of blood vessel wall, the illness developed into the most serious form, hepatic-spleen type schistosomiasis [22,23]. Liver fibrosis is a reversible wound repair process [24,25]. Previous studies showed that liver fibrosis can further aggravate if no interference treatment is conducted [26]. Since 2005, a rescue policy for advanced schistosomiasis patients has been implemented in China [27], which not only improved the life quality of patients but also eased their economic burden. However, the total number of advanced schistosomiasis cases in China was maintained at 29,407–30,170 during 2013–2019, with nearly 1000 advanced case deaths and about 1000 new advanced cases reported annually [5,28,29,30,31]. Scarce of sensitive diagnosis and failure of timely anti-fibrosis treatment might contribute to the newly reported advanced cases [7,32]. 

Being a non-invasive method, ultrasonography is considered to be a complementary tool for diagnosis of intestinal schistosomiasis, particularly for assessing the liver damage caused by chronic infection. For *S. japonicum*, adult females produce 10 times more eggs per day than *S. mansoni*, and the eggs often deposit in clusters, thus exacerbating the severity of *S. japonicum* morbidity relative to other schistosome species and contributing to the unique fibrotic pattern. In our study, the liver fibrosis assessed by ultrasound scan remained stable in nearly 50% of the participants, but deteriorated to higher grade in more than 20% of participants. It is indicated that although patients received deworming treatment, the pathological harm caused by schistosomes to the human body did not stop. Those patients who got infected and cured by etiologic treatment are meaningless in terms of “transmission”, but the pathological damage they received is still undergoing and developing. Therefore, due to the risk of developing “schistosomiasis liver fibrosis”, people who were infected with schistosomes should be given more attention under the strategy of “Healthy China, Universal Health”. Conversely, the study also found that liver fibrosis was alleviated in some participants with a recovery rate of 22.26%, suggesting the importance of timely treatment and interventions against liver fibrosis. 

Several studies indicated that the immune response of human body, exposure frequency and reinfection intervals were involved in the occurrence of liver fibrosis. Other factors such as host genetic susceptibility, malnutrition, repeated infections and treatments, co-infections with hepatitis, tuberculosis, and typhoid etc., are also risk factors [9,10]. Receiving treatment and improved education are considered as protective factors [8]. Housework such as laundry, washing vegetables [33] are the main ways that women are exposed to fresh water, which can be easily changed by health education [34]. The liver fibrosis among young patients is mostly at an early stage, which means their liver fibrosis is more reversible [35]. Due to frequent exposure to contaminated water, fishermen are prone to repeated infections or failure of timely treatment, which can result in further damage to their liver, spleen, and other organs [14].

In our study, the prevalence of schistosomiasis decreased dramatically, and no new infection occurred in 2019, but the ratio of liver fibrosis with higher grade increased from 1995 to 2019. Logistic regression analysis revealed that the factors affecting the development of schistosomiasis liver fibrosis include gender, age, occupation, and anti-fibrosis treatment. In particular, anti-liver fibrosis therapy can effectively reduce the deterioration of the liver fibrosis. The result has also been verified by Sobhy M et al. [36]. To prevent the deterioration of hepatic fibrosis and reduce the infection related morbidity caused by schistosomiasis, surveillance should be strengthened particularly in those areas that had prior infection. Continuous follow-up and screening on the determined schistosomiasis cases and population at high risk of infection by ultrasound examination could improve the diagnosis of intestinal schistosomiasis and understand the dynamics of hepatic fibrosis to benefit conducting interventions to reduce disease burden.

There are some limitations in our study. One is that the study was only conducted in one village of Jiangxi province with a small sample. Whether the findings reflect other endemic areas needs further verification. Another is that the data obtained from participants was received using a questionnaire survey that might be affected by memory biases. We recommend conducting large scale follow-up studies to assess the liver morbidity of population and understand its influence factors in endemic areas at different stages of schistosomiasis control and elimination in China.

## 4. Materials and Methods

### 4.1. Study Area and Cohort

Xinhua village, Lushan city of Jiangxi Province, was selected as the study site. The village is a typical lake and marshland endemic village of schistosomiasis located downstream of the Ganjiang River, northern bank of Poyang Lake. The prevalence of schistosomiasis in this village was very high but decreased to less than 1% by 2015. Fishing, laundry, and wading were the main ways that residents became infected. The area of snail habitats was 1,697,500 m^2^ at an altitude from 11.28 to 13.00 m. Residents at the age of 5 and above in this village were selected to participate in our study. During the study period, no infection with other parasites, such as *Fasciola hepatica*, *Capillaria hepatica*, *Clonorchis sinensis*, and *Echinococcus granulosus*, etc., that may cause liver damage were detected in this study area.

### 4.2. Parasitological Survey

Stool examination was conducted annually to understand the infection status of schistosomiasis among residents during 1995 and 2019. Considering the sensitivity and accuracy of routine pathogenic examinations declined with the infection intensity [26,27,28], the parasitological methods and the number of stool samples collected from residents varied in different periods. Specifically, from 1995 to 2004, each participant provided 1 stool sample, and 3 slides were prepared following the instruction of the modified Kato–Katz thick smear procedure. From 2004 to 2007, 2 stool samples were collected from each participant at intervals of 3–5 days and 3 slides for each sample were prepared for testing. From 2008 to 2019 (except 2014), each participant provided 3 stool samples, examined by the Kato–Katz thick smear methods and the miracidia hatching technique in parallel [37]. The infection was confirmed if eggs or miracidia of *Schistosoma japonicum* were detected in stool samples. The infected residents were treated with praziquantel at dose of 40 mg/kg body weight, and their feces were collected and re-examined one month after treatment. The person who tested positive received a second treatment to ensure a cure.

### 4.3. Ultrasound Examination

A portable ultrasound apparatus (Hitachi EUB-40 with a curvilinear probe 3.5 MHz, Hitachi, Japan) was used for liver ultrasound examination in 1995 and 2019, respectively, by an experienced ultrasonographer. Unique to *S. japonicum* infection is parenchymal fibrosis, a network pattern that is often described as fish scale or tortoise shell-like. Hepatic fibrosis grading was carried out according to the standard of practical ultrasound diagnostic guidelines (1990) developed by the World Health Organization/Tropical Disease Research Organization. The liver fibrosis was graded from 0 to 4: (1) Grade 0: normal and thicker light spot type, normal liver sonogram or only thick liver parenchyma echo; (2) Grade I: focal echoes in the liver parenchyma are scattered without clear boundary; (3) Grade II: fish-scale and cobweb type, a few focal echo density areas < 20 mm; (4) Grade III: echo density bands form a continuous network, multifocal echo areas > 20 mm, masses with central fibrosis [38] (Figure 3).

### 4.4. Questionnaire Survey

A uniformly designed questionnaire was used to investigate all respondents who received ultrasound examination in 2019. Information included gender, age, occupation, education level, history of water contact, and treatment (treatment time, number of treatments, etc.), compliance with treatment, etc., were collected through the questionnaire survey. Anti-fibrosis treatment consists of liver function improvement, liver stiffness, and liver fibrosis reduction by using polyene phosphatidylcholine, reduced glutathione, or hepatocyte growth factor, etc., along with supplement from liver protection tablets for routine liver protection.

### 4.5. Quality Control

The stool samples collected from participants should be fresh and sufficient, otherwise a second collection will required. The microscopic examination was carried out by 3 skilled technicians back-to-back 24 h after the smears were prepared. The positive results were confirmed by 2 technicians independently. The ultrasound examination was performed by the same operator with the same ultrasound apparatus in 1995 and 2019, respectively, without knowing schistosome infection status. All the investigators had received systematic training before implementing the questionnaire survey.

### 4.6. Data Management and Statistics

All data analysis and processing were carried out using “Statistical Product and Service Solutions” (SPSS) V 20.00 software package (IBM, Armonk, NY, USA). For variables of age and treatment times, we converted them into categorical variables in order to analyze the factors that affect the progress of liver fibrosis. For example, age at 38 years old was set as the age grouping standard (median of frequency distribution calculated in 1995 was 37.8 years old), and 8 times of treatments was regarded as the grouping standard (the median value of frequency distribution calculation was 8). The univariate analysis with χ^2^ test and multivariate logistic analysis by the “stepwise backward” method were conducted to explore the variables influencing the changes of liver fibrosis [39]. The final report was presented with odds ratios (ORs), 95% confidence intervals (95% CI) with a *p* value less than 0.05 set as statistical significance.

The evaluation on the change of liver fibrosis is reflected by two indicators: “recovery rate” and “deteriorating rate”. Recovery means the grading of liver fibrosis examined in 2019 was lower than that in 1995. Deterioration means the grading of liver fibrosis examined in 2019 was higher than that of 1995. The recovery rate and deteriorating rate were calculated using the following formulas:$$\mathrm{Recovery}\;\mathrm{rate}\;(\%)=\frac{The\;number\;of\;people\;with\;milder\;liver\;fibrosis}{The\;number\;of\;people\;received\;ultrasound\;examination}\times 100\%$$
$$\mathrm{Deteriorating}\;\mathrm{rate}\;(\%)=\frac{The\;number\;of\;people\;with\;severer\;liver\;fibrosis}{The\;number\;of\;people\;received\;ultrasound\;examination}\times 100\%$$

## 5. Conclusions

In summary, although the prevalence of schistosomiasis has been effectively controlled, the aggravation of liver fibrosis still occurs in some patients. It is recommended to raise the awareness of people to actively receive schistosomiasis examination and treatment after they had history of water contact in schistosomiasis endemic areas. In particular, liver protection/anti-liver fibrosis treatment for schistosomiasis patients should be strengthened to slow down the development of liver fibrosis in residents. Continuous follow-up and screening on the confirmed cases and population at high risk of schistosome infection by ultrasound examination could improve the diagnosis of intestinal schistosomiasis and understand the dynamics of hepatic fibrosis to benefit conducting interventions to reduce disease burden.

## Figures and Tables

**Figure 1 pathogens-10-01532-f001:**
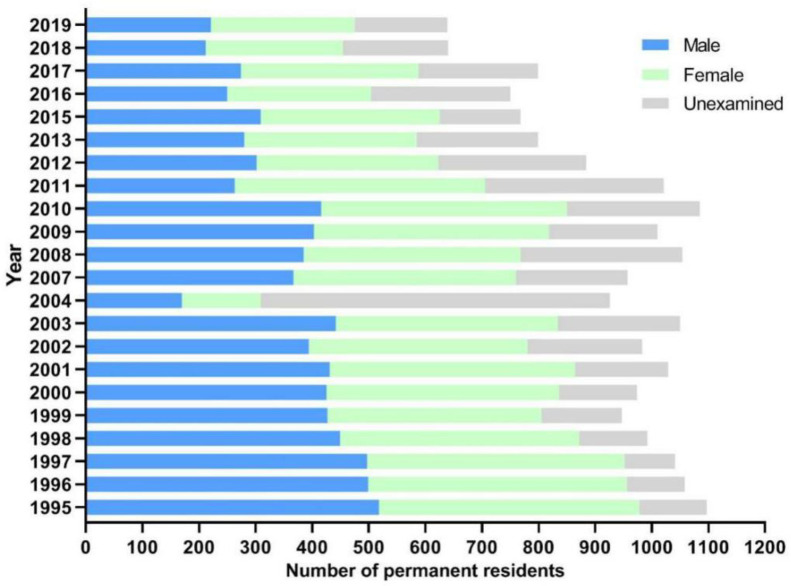
Basic information of participants in Xinhua village from 1995 to 2019.

**Figure 2 pathogens-10-01532-f002:**
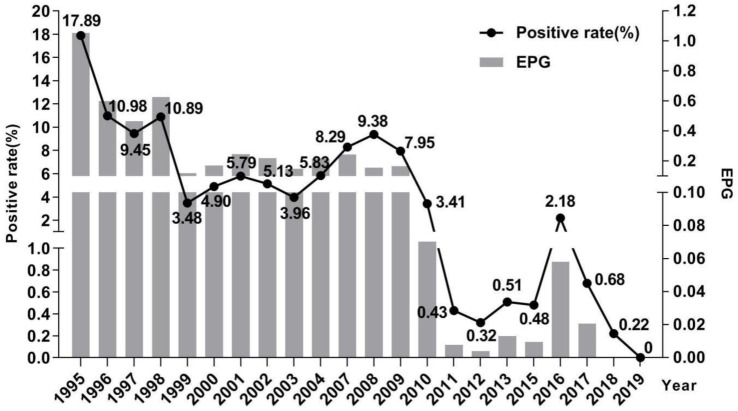
Schistosomiasis positive rate and EPG by fecal examination in residents of Xinhua village during 1995–2019.

**Figure 3 pathogens-10-01532-f003:**
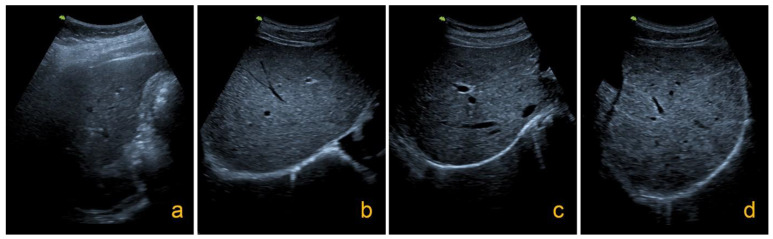
Different grading images of liver fibrosis in schistosomiasis ((**a**) = Grade 0; (**b**) = Grade I; (**c**) = Grade II; (**d**) = Grade III).

**Table 1 pathogens-10-01532-t001:** Comparison of the grading of schistosomiasis hepatic fibrosis between 1995 and 2019.

1995	2019				
	Grade 0	Grade I	Grade II	Grade III	Subtotal
Grade 0	92	26	3	1	122
Grade I	53	38	30	10	131
Grade II	1	7	7	16	31
Grade III	0	1	3	4	8
Total	146	72	43	31	292

**Table 2 pathogens-10-01532-t002:** Univariate analysis of the evolution of liver fibrosis in schistosomiasis.

Factors		Investigated Number	Recovery (*n* = 65)					Deterioration (*n* = 86)				
			No.	χ^2^	OR	95% CI	*p* Value	No.	χ^2^	OR	95% CI	*p* Value
Gender	Male	146	27	2.395	0.645	0.369–1.127	0.122	59	16.878	2.989	1.754–5.092	0.000
	Female	146	38					27				
Age group	<38	149	30	0.795	1.285	0.740–2.234	0.373	31	10.948	2.379	1.415–4.000	0.001
	≥38	143	35					55				
Occupation	Fishermen	66	8	5.066	0.409	0.184–0.908	0.024	29	8.614	2.324	1.312–4.115	0.003
	Non-fishermen	226	57					57				
Education	Illiterate	125	34	7.689			0.104	35	2.899			0.575
	Primary school	106	25					36				
	Junior high school	46	4					10				
	Senior high school	15	2					5				
Egg positive	Yes	139	35	0.07	0.928	0.534–1.613	0.791	37	4.295	1.707	1.027–2.837	0.038
	No	153	30					49				
Water contact	Frequently	137	30	0.02	0.961	0.553–1.671	0.889	40	0.008	0.977	0.590–1.618	0.928
	Infrequently/No	155	35					46				
No. treatment	<8	145	39	3.578	0.584	0.333–1.023	0.059	31	9.034	2.198	1.309–3.694	0.003
	≥8	147	26					55				
Active medication	Yes	29	4	1.334	0.530	0.177–1.582	0.248	9	0.039	1.087	0.474–2.494	0.844
	No	263	61					77				
Anti-fibrosis treatment	Yes	150	41	4.587	1.849	1.049–3.261	0.032	34	6.835	0.507	0.304–0.847	0.009
	No	142	24					52				

**Table 3 pathogens-10-01532-t003:** Multivariate logistic regression analysis of the evolution of liver fibrosis in schistosomiasis.

Factors	Recovery			Deterioration		
	OR	95% CI	*p* Value	OR	95% CI	*p* Value
Gender	0.964	0.471–1.974	0.921	2.735	1.523–4.910	0.001
Age group				2.986	1.668–5.347	0.000
Profession	0.480	0.205–1.122	0.090	2.416	1.229–4.749	0.010
Education			0.322			
Egg positive				1.411	0.762–2.611	0.273
No. treatment	0.568	0.304–1.060	0.076	1.651	0.895–3.045	0.108
Anti-fibrosis treatment	2.277	1.249–4.151	0.007	0.302	0.165–0.553	0.000

## Data Availability

The data that support the figures within this paper and other findings of this study are available from the corresponding authors upon reasonable request.

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
