# Peer review of "The Dynamics of Hepatic Fibrosis Related to Schistosomiasis and Its Risk Factors in a Cohort of China"

_pathogens, 2021, doi:10.3390/pathogens10121532_

Round 1

Reviewer 1 Report

Estimated Authors of the paper "The dynamics of hepatic fibrosis related to schistosomiasis and its risk factors in a cohort of China",

I've read your article with great interest. Schistosomiasis is among the neglected tropical disease one of the main cause of morbidity (but also, from a longer point of view, of mortality) of a large share of the human population worldwide. The efforts paid by PRC to reduce the burden of schistosomiasis in the Chinese population is well known and somewhat revered by professionals involved in the management of infectious diseases.

The present report, through the analysis of the residents (i.e. subjects dealing in the area for at least 5 years) in a schistosomiasis endemic village were examined in order to identify signs of schistosomiasis infection every year between 1995-2019 by the modified Kato-Katz thick smear procedure and the egg hatching method. Authors have eventually identify factors associated with recovery and development of liver fibrosis (regarding the latter: gender, age, profession having NOT received anti-fibrotic treatment). Such results are quite interesting for professionals working with NTD.

Unfortunately, from my point of view, the paper is also affected by two main shortcomings that Authors should fix before a full publication.

First and foremost: the overall quality of the English text MUST benefit from an extensive revision. Some instances of jargon / typos that affect this paper:

After more than seven decades’ endeavor, remarkable achievements had gained in schistosomiasis control in China --> did you mean something alike: "After more than 7 decades of efforts, PRC has gained a substantial control of schistosomiasis in large areas of Mainland China"?

It was reported that nearly 20% patients will develop to liver fibrosis --> Previous reports suggest that around 20% of patients develop liver fibrosis

In China, about 1000 new advanced schistosomiasis cases were developed and diagnosed annually, while some of them presented in endemic areas where the transmission of schistosomiasis had been interrupted many years ago --> In China, about 1000 new advanced schistosomiasis cases are annually diagnosed; while some of them are reported from endemic areas where the transmission of schistosomiasis was interrupted many years ago ... 

There were 292 people received liver ultrasound and fecal examination both in 1995 and 2019 --> There were 292 people receiving liver ultrasound and fecal examination both in 1995 and 2019

etc.

Please be aware that the aforementioned are only some examples. Please double check your text before a new submission.

Second, the methods section is substantially well written (even though not exempt from typos and jargons) BUT it should moved before the results section, as in the current position the reader may fail to understand the actual content of the paper.

Author Response

Dear Reviewers:

Thank you for your interest in our manuscript entitled “The dynamics of hepatic fibrosis related to schistosomiasis and its risk factors in a cohort of China”. We have studies comments carefully and have made correction which we hope meet the final approval. The alterations in the text are highlighted in the paper as requested. The main corrections in the paper and the responses to the reviewer’s comments are as following:

  1. First and foremost: the overall quality of the English text MUST benefit from an extensive revision. Some instances of jargon / typos that affect this paper:

Response: Thank you very much for your com ments. We have invited experts in this field helping us improved the language and modified the inappropriate expression in our resubmitted manuscript.

  1. Second, the methods section is substantially well written (even though not exempt from typos and jargons) BUT it should moved before the results section, as in the current position the reader may fail to understand the actual content of the paper.

Response: Thank you for your positive comments. However, the structure of manuscript submitted to Pathogen is fixed and we settled each part following the instruction of the editorial office of the journal of Pathogen. So we didn’t adjust the format of our resubmitted manuscript.

Reviewer 2 Report

The authors report on hepatic fibrosis in a Chinese cohort as recorded in 1995 and 2017 and associate it with the course of the incidence rates of schistosomiasis. Although the idea is interesting, I have severe concerns regarding the methodical approach, so I cannot recommend for publishing the manuscript as it stands.

1.) The methods used for the diagnosis of schistosomiasis were relevantly changed in the course of the study, resulting in higher sensitive approaches in 2017 and making it likely that a higher proportion was overlooked in 1995. Those procedural changes are a source of bias which is difficult to control and it is not obvious how the authors have tried to control it at all.

2.) Although schistosomiasis can be associated with liver fibrosis, multiple other diseases or habits can cause this medical condition as well. Without a thorough assessment of such potential sources of bias and just based on the few additional assessed factors from tables 2 and 3, the results are difficult to interpret.

3.) Although the authors mentioned that lacking consent of participants of the cohort might have accounted for varying proportions of unexamined individuals over the years, this hardly explains very high discrepancies as the overwhelming unexamined majority for the year 2004 (as indicated in figure 1). Such shifts suggest systematic bias.

4.) Language quality does not meat the requirements for a peer-reviewed international publication. Proofreading by a native speaker with experience in scientific editing is recommended.

Both the epidemiological data on schistosomiasis incidence and on prevalence of liver fibrosis are interesting, however, the provided evidence for the claimed causal association between both is scarce.

Author Response

Dear Reviewers:

Thank you for your interest in our manuscript entitled “The dynamics of hepatic fibrosis related to schistosomiasis and its risk factors in a cohort of China”. We have studies comments carefully and have made correction which we hope meet the final approval. The alterations in the text are highlighted in the paper as requested. The main corrections in the paper and the responses to the reviewer’s comments are as following:

  1. The methods used for the diagnosis of schistosomiasis were relevantly changed in the course of the study, resulting in higher sensitive approaches in 2017 and making it likely that a higher proportion was overlooked in 1995. Those procedural changes are a source of bias which is difficult to control and it is not obvious how the authors have tried to control it at all.

Response: Thank you for your comments. Yes, the parasitological methods and slides prepared changed during our research. As we explained in the resubmitted manuscript, the endemicity of schistosomiasis changed greatly during 25 years covered by our study. At the early stage of the study, the infection rate and infection intensity were high. The sensitivity of the Kato-Katz's method is quite good to examine 3 slides for one stool sample. With the progress of research, the prevalence and infection intensity of schistosomiasis in residents both declined. To improve the sensitivity of diagnostic during the middle and late stages of the study, the diagnostic approach changed but it does not mean that the infected residents was missed at the early stage of the study as the prevalence was quite high. In addition, the data reflecting the prevalence of schistosomiasis only provides the general background for the study. In our research, we mainly focused on analyzing the ultrasonagraphy data which obtained from uniformly ultrasound scan. The examination methods of liver fibrosis, grading criteria for liver fibrosis and even examiners remain same between 1995 and 2019.

  1. Although schistosomiasis can be associated with liver fibrosis, multiple other diseases or habits can cause this medical condition as well. Without a thorough assessment of such potential sources of bias and just based on the few additional assessed factors from tables 2 and 3, the results are difficult to interpret.

Response: We agree with your comments that there are other pathogens and diseases could cause liver fibrosis. Indeed, when conducting our research, except general information of participants, the questionnaire designed also collected the questions about the history of other diseases, particularly liver disease caused by other factors. We also detected other parasites which may cause liver damage, such as Fasciola hepatica, Capillaria hepatica, Clonorchis sinensis, etc., at the same time we detected schistosoma japonium when read kato-katz's thick smear slides. And no infection of other liver damage related parasite was found. Thus, we've made a supplementary in "4.1 Study area and cohort" in the method section to explain the information discussed above. In addition, we supplement some sentence to explain the specific fibrosis pattern caused by Schistosoma japonicum as a fish scale or tortoise shell-like, which could be distinguished from other liver fibrosis caused by viral hepatitis, alcoholic liver, and fatty liver, in our resubmitted manuscript.

  1. Although the authors mentioned that lacking consent of participants of the cohort might have accounted for varying proportions of unexamined individuals over the years, this hardly explains very high discrepancies as the overwhelming unexamined majority for the year 2004 (as indicated in figure 1). Such shifts suggest systematic bias.

Response: Thank you very much for your comments. All the data reflecting the changes of prevalence of schistosomiasis in residents only provide general background to reveal that the prevalence decreased dramatically, but didn’t impact the following univariate and multivariate analysis on the liver fibrosis. If the reviewer insist, we can delete the data of year 2004, this  would not impact on the main result of this manuscript.

  1. Language quality does not meat the requirements for a peer-reviewed international publication. Proofreading by a native speaker with experience in scientific editing is recommended.

Response: We are sorry for our poor writing skills. We have invite two experienced experts to improve the language quality and hope the resubmitted version could meet the requirements of this journal.

Reviewer 3 Report

The manuscript “The dynamics of hepatic fibrosis related to schistosomiasis and its risk factors in a cohort of China” reports the results of a study on schistosomiasis in the residents of a village during 25 years of follow up with coprological examinations and liver fibrosis evaluation at the start and the end of the study.

The concept of the study is interesting and the duration of follow up is impressive. However, there are several points of the survey that need clarification or justification.

The authors should clearly state why they selected to apply the examination to only one sample at the beginning of the study and gradually increase the examinations (number of samples and methods) towards the end of the study.

Were the patients examined for other conditions that could affect the liver? There are numerous pathologies associated with liver fibrosis. How were they excluded? Could other pathologies co-exist with parasitism in some cases?

What does “Anti-fibrosis treatment” consist of?

Cases of re-infections are not reported in the population examined. Were there such cases?

Lines 174 and 176. What is the difference between reinfection and repeated infection?

Line 186. What does “etc” stand for here? What other factors affected the development of schistosomiasis liver fibrosis?

Regarding the presentation of the results, there are some confounding data. For example, in lines 33-34: “… anti-liver fibrosis treatment was protective factor of liver fibrosis, but- ……. anti-liver fibrosis treatment were correlated with development of liver fibrosis”. Are the authors here reporting the same conclusion twice? Are the results suggesting that when the patient receives anti-liver fibrosis treatment the liver fibrosis is not developing/is recovering/is developing slower?

The results should be presented more clearly. The tables should be supported with the corresponding text. For example, mentioning that gender is a factor associated with the development of schistosomiasis liver fibrosis is not enough. Which gender is more prone to this condition should be mentioned in the Results and discussed in the Discussion. The same is valid for the rest of the findings.

I am sorry, but I cannot understand Table 1.

Finally, the English language needs thorough editing. In many places, it is difficult to understand what the authors mean to say.

Author Response

Dear Reviewers:

Thank you for your interest in our manuscript entitled “The dynamics of hepatic fibrosis related to schistosomiasis and its risk factors in a cohort of China”. We have studies comments carefully and have made correction which we hope meet the final approval. The alterations in the text are highlighted in the paper as requested. The main corrections in the paper and the responses to the reviewer’s comments are as following:

  1. The authors should clearly state why they selected to apply the examination to only one sample at the beginning of the study and gradually increase the examinations (number of samples and methods) towards the end of the study.

Response: Thank you very much for your comments. We have supplemented the detailed information of prevalence survey and explained the reason of changing the parasitological methods during the years covered in our study in the part of method 4.2 in our resubmitted manuscript.

  1. Were the patients examined for other conditions that could affect the liver? There are numerous pathologies associated with liver fibrosis. How were they excluded? Could other pathologies co-exist with parasitism in some cases?

Response: Yes, there are many other pathogens or risks associated with liver fibrosis. As we explained in previous question asked by another reviewer, we supplement the information of questionnaire we collected and also the stool examination didn’t find any infection which may cause liver damage, such as Fasciola hepatica, Capillaria hepatica, Clonorchis sinensis, and Echinococcus granulosus, etc. Thus, we've made a supplementary in "4.1 Study area and cohort" in the method section to explain the information discussed above. In addition, we supplement some sentence to explain the specific fibrosis pattern caused by Schistosoma japonicum as a fish scale or tortoise shell-like, which could be distinguished from other liver fibrosis caused by viral hepatitis, alcoholic liver, and fatty liver, in our resubmitted manuscript. That’s why we didn’t mention the exclusion criteria in the paper.

  1. What does “Anti-fibrosis treatment” consist of?

Response: Anti-fibrosis treatment consist of liver stiffness reduction, liver function improvement, and liver fibrosis reduction with polyene phosphatidylcholine, reduced glutathione or hepatocyte growth factor, etc., and routine liver protection with liver protection tablets. We have made a supplementary in "4.4 Questionnaire survey" in the method section in our revised manuscript.

  1. Cases of re-infections are not reported in the population examined. Were there such cases?

Response: Yes, there were reinfection occurred during the research period. Regarding the data of re-infections has been published in another paper, here we added a sentence briefly as following in 2.2 Changes of schistosomiasis prevalence in participants" in the results section as: The re-infection rates among residents ranged from 4.29 – 38.46% during 1996 to 2011.Sine 2011, reinfection only occurred in 2012 and 2017”.

  1. Lines 174 and 176. What is the difference between reinfection and repeated infection?

Response: Reinfection refers to people got infection but cured after treatment, and infect again with schistosomes in the next transmission season. Repeated infection refers to multiple infection or exposure to schistosomes during a period of observation with or without treatment.

  1. Line 186. What does “etc” stand for here? What other factors affected the development of schistosomiasis liver fibrosis?

Response: "etc." has been deleted in revised manuscript.

  1. Regarding the presentation of the results, there are some confounding data. For example, in lines 33-34: “… anti-liver fibrosis treatment was protective factor of liver fibrosis, but- ……. anti-liver fibrosis treatment were correlated with development of liver fibrosis”. Are the authors here reporting the same conclusion twice? Are the results suggesting that when the patient receives anti-liver fibrosis treatment the liver fibrosis is not developing/is recovering/is developing slower?

Response: This part had been revised as “The results showed that anti-fibrosis treatment is the protective factor against schistosomiasis hepatic fibrosis. Male, people at age of 38 or above, fishermen and people who didn’t received anti-fibrosis treatment are higher risk groups of liver fibrosis development”. The results of these analyses had been discussed in the discussion section.

  1. The results should be presented more clearly. The tables should be supported with the corresponding text. For example, mentioning that gender is a factor associated with the development of schistosomiasis liver fibrosis is not enough. Which gender is more prone to this condition should be mentioned in the Results and discussed in the Discussion. The same is valid for the rest of the findings.

Response: Thank you for your suggestion. We have revised the part of results according to your suggestion. Related analysis and discussion were also revised in the part of discussion in our resubmitted manuscript.

  1. I am sorry, but I cannot understand Table 1.

Response: Table 1 shows the grading of liver fibrosis assessed by liver ultrasound in 1995 and 2019. The leftmost column lists the grade of 1995, the top line lists the grade of 2019. The number in each shell is the frequency of liver fibrosis with grade ## in both 1995 and 2019.

  1. Finally, the English language needs thorough editing. In many places, it is difficult to understand what the authors mean to say.

Response: We are sorry for our poor writing skills. We have invite two experienced experts to improve the language quality through the whole manuscript. 

Round 2

Reviewer 2 Report

I appreciate the authors efforts to address the limitations of the study. Regarding the "specific" fibrosis pattern caused by Schistosoma japonicum, which the authors describe as “a fish scale or tortoise shell-like” appearance, there are most likely specificity estimations available, aren’t they? If yes, those estimations should be included in the calculation, if no, this limitation should be frankly discussed.

Author Response

Dear Reviewers:

Thank you for your interest in our manuscript entitled "The dynamics of hepatic fibrosis related to schistosomiasis and its risk factors in a cohort of China". According to your comments, we revised our manuscript and invite a native speaker to improve the language quality. The modifications were highlighted in the resubmitted manuscript.

  1. I appreciate the authors efforts to address the limitations of the study. Regarding the "specific" fibrosis pattern caused by Schistosoma japonicum, which the authors describe as “a fish scale or tortoise shell-like” appearance, there are most likely specificity estimations available, aren’t they? If yes, those estimations should be included in the calculation, if no, this limitation should be frankly discussed.

Response: Thank you for your comments. Yes, the imaging of liver fibrosis caused by infection of S. japonicum is specific as we described in section 4.3. For fibrosis, “a fish scale or tortoise shell-like”was only used for qualitative analysis to determine whether the fibrosis was caused by the infection of S. japonicum. For grading, the standards were described in the manuscript according to "Ultrasound in schistosomiasis: a practical guide to the standardized use of ultrasonography for the assessment of schistosomiasis-related morbidity (2000)". In addition, we added one paragraph to analyze the limitations in this study as followings in our resubmitted manuscript: “There are some limitations in our study. One is that the study was only conducted in one village of Jiangxi province with a small sample. Whether the founding reflected other endemic areas need further verification. Another one is that the data obtained from participants received questionnaire survey and twice ultrasound scan might be affected by memory biases.

We recommend to conduct large scale follow-up studies to assess the liver morbidity of population and understand its influence factors in endemic areas at different stages of schistosomiasis control and elimination in China.”

We revised seriously the English language again.

Reviewer 3 Report

The authors have significantly improved the presentation of their work. Please see my specific comments below:

  1. Line 21. Please correct emerges to read emerge
  2. Line 23. Please add the word “old” after “years”
  3. Lines 95-97. Following the author's reply to this reviewer: Please refer to the publication in which the data of reinfections in the same population sample has been published and cite the paper.
  4. Table 1. I am sorry but I still do not understand Table 1, no matter how much I have tried. However, if the Editor believes it is correct and comprehensible, I will not insist.
  5. Line 218. Please delete the words “prevalence and” because it is only the reduced intensity of infection (if it is depicted in the eggs per gram) that would influence the prevalence determined by a method (depending on its sensitivity) while the actual prevalence would not have any effect on this level.
  6. Correct to italics all scientific names of parasites

Author Response

Dear Reviewers:

Thank you for your interest in our manuscript entitled "The dynamics of hepatic fibrosis related to schistosomiasis and its risk factors in a cohort of China". According to your comments, we revised our manuscript and invite a native speaker to improve the language quality. The modifications were highlighted in the resubmitted manuscript.

  1. Line 21. Please correct emerges to read emerge.

Response: Thank you. It had been corrected.

  1. Line 23. Please add the word "old" after "years".

Response: Thank you. The word "old" had been added in the manuscript.

  1. Lines 95-97. Following the author's reply to this reviewer: Please refer to the publication in which the data of reinfections in the same population sample has been published and cite the paper.

Response: Thank you. The paper had been included in the references.

  1. Table 1. I am sorry but I still do not understand Table 1, no matter how much I have tried. However, if the Editor believes it is correct and comprehensible, I will not insist.

Response: Thank you for your comments. Table 1 is a crosstable used to compare the frequency of liver fibrosis with different grades assessed in 1995 and 2019 respectively. A typical crosstable is fourfold table, but here is 4×4 fold table as the liver fibrosis was classified to four categories. The recovery rate and deteriorating rate were calculated based on this table. For example, there were 122 people in Grade 0 in 1995, in 2019, 92 people still classified as Grade 0, 26 developed to Grade I, 3 developed to Grade II and 1 people developed to Grade III. So, here, liver fibrosis deteriorated in 30 people (26+3+1) during the study period. Similarly, there were total 131 people in Grade I in 1995, among of them, 53 recovered from Grade I to Grade 0, 38 people who are still in Grade I, 30 people developed to Grade II, 10 people developed to Grade III. Thus liver fibrosis recovered in 53 people and deteriorated in 40 people. Finally, from table1 the liver fibrosis deteriorated in 86 (40+30+16) people, recovered in 65 (53+8+4) people. Therefore, from table1, hepatic fibrosis was developed severely in 29.45% (86/292) of participants but reversed in 22.26% (65(=53+8+4)/292) of residents.

Table 1. Comparison of the grading of schistosomiasis hepatic fibrosis between 1995 and 2019.

1995

2019

Grade 0

Grade I

Grade II

Grade III

Subtotal

Grade 0

92

26

3

1

122

Grade I

53

38

30

10

131

Grade II

1

7

7

16

31

Grade III

0

1

3

4

8

Total

146

72

43

31

292

  1. Line 218. Please delete the words "prevalence and" because it is only the reduced intensity of infection (if it is depicted in the eggs per gram) that would influence the prevalence determined by a method (depending on its sensitivity) while the actual prevalence would not have any effect on this level.

Response: Thank you for your suggestion. The words "prevalence and" had been deleted.

  1. Correct to italics all scientific names of parasites

Response: Thank you for your suggestion. All scientific names of parasites had been corrected to italics.

We revised seriously the English language again.